# The Predictive Value of Systemic Immune-Inflammation Index on Bladder Recurrence on Upper Tract Urothelial Carcinoma Outcomes after Radical Nephroureterectomy

**DOI:** 10.3390/jcm10225273

**Published:** 2021-11-12

**Authors:** Tsu-Ming Chien, Ching-Chia Li, Yen-Man Lu, Yii-Her Chou, Hsueh-Wei Chang, Wen-Jeng Wu

**Affiliations:** 1Graduate Institute of Clinical Medicine, College of Medicine, Kaohsiung Medical University, Kaohsiung 80756, Taiwan; u108801005@kmu.edu.tw (T.-M.C.); yihech@kmu.edu.tw (Y.-H.C.); 2Department of Urology, Kaohsiung Medical University Hospital, Kaohsiung 80756, Taiwan; ccli1010@hotmail.com; 3Department of Urology, Faculty of Medicine, College of Medicine, Kaohsiung Medical University, Kaohsiung 80756, Taiwan; yimen1980@hotmail.com; 4Department of Urology, Kaohsiung Municipal Ta-Tung Hospital, Kaohsiung 80145, Taiwan; 5Department of Biomedical Science and Environmental Biology, College of Life Science, Kaohsiung Medical University, Kaohsiung 80708, Taiwan; 6Cancer Center, Kaohsiung Medical University Hospital, Kaohsiung 80708, Taiwan; 7Center for Cancer Research, Kaohsiung Medical University, Kaohsiung 80708, Taiwan

**Keywords:** upper tract urothelial carcinoma, bladder recurrence, survival, platelet-lymphocyte, neutrophile-lymphocyte, systemic immune-inflammation index

## Abstract

Background: This study aimed to assess the prognostic significance of pre-treatment lymphocyte-related systemic inflammatory biomarkers in upper tract urothelial carcinoma (UTUC) patients. Methods: This study included non-metastatic UTUC patients treated at our hospital between 2001 and 2013. The receiver operating characteristic curve was used to obtain the optimal neutrophile-lymphocyte ratio (NLR), platelet-lymphocyte ratio (PLR), and systemic immune-inflammation index (SII). Multivariate logistic regression was performed to investigate the relationship between NLR, PLR, and SII and clinical pathologic characteristics. The Kaplan–Meier method was used to calculate the metastasis-free survival (MFS), cancer-specific survival (CSS), and bladder recurrence-free survival (BRFS), and the log-rank test was used to compare the survival rate. Results: Overall, 376 patients were included in the current study. An elevated SII was associated with symptomatic hydronephrosis, bladder cancer history, advanced pathologic tumor stage, lymph node invasion, adjuvant chemotherapy and concomitant carcinoma in situ (CIS); high NLR was associated with older age, symptomatic hydronephrosis, hemodialysis status, anemia, multifocal tumor, advanced pathologic tumor stage, and adjuvant chemotherapy; and high PLR was associated with older age, anemia, advanced pathologic tumor stage, and adjuvant chemotherapy. The Kaplan–Meier analysis indicated that patients exhibiting higher NLR, PLR, and SII showed significantly poor MFS and CSS rates. Only high SII showed significantly worse BRFS rates. Conclusions: The NLR, PLR, and SII were independent predictive factors for both MFS and CSS in UTUC patients. Among the factors, only elevated SII can predict bladder recurrence. Therefore, the patients might need close bladder monitoring during the follow-up.

## 1. Introduction

Urothelial carcinoma (UC) is the most common malignancy and includes UC of the urinary bladder (UBUC) and upper tract urothelial carcinoma (UTUC). UTUCs are uncommon, accounting for only 5–10% of UCs [1], with an estimated annual incidence in Western countries of almost two cases per 100,000 inhabitants [1,2]. During initial diagnosis, almost 60% of UTUCs were invasive, which is higher than that in UBUC [3]. Several studies have attempted to identify prognostic factors in patients with UTUC [3,4,5,6]. These models are based on age, tumor grade, pathologic tumor grade, lymph node metastasis, lymph vascular invasion, tumor multifocality, and tumor architecture [3,4,5,6]. With the development of current genomics studies of UTUC [7], physicians can integrate the clinical pathologic characteristics with genetic and molecular subtyping of UTUC. Knowing the different genotype and phenotype features can lay the groundwork for a deeper understanding of UTUC biology [8]. This information is extremely beneficial in terms of personalized treatments regarding postoperative neoadjuvant chemotherapy and follow-up planning. However, most of these factors are obtained only after surgical removal of the tissue. The considerable improvement of imaging studies, such as computed tomography or magnetic resonance imaging, can provide valuable information in patient counseling and treatment planning [9]; however, its accuracy does not meet the standard for individual treatment decision-making. Therefore, preoperative data to predict recurrence and survival is urgently needed.

An increasing number of studies have proved that inflammatory reaction in the tumor microenvironment is associated with tumor development and progression [10,11]. As inflammation can stimulate leukocytosis, neutrophilia, and thrombocytosis, various blood-based inflammatory markers, such as neutrophile-to-lymphocyte ratio (NLR), platelet-to-lymphocyte ratio (PLR), and monocyte-to-lymphocyte ratio (MLR), have been investigated to improve prognostic tools for risk stratification and outcomes in UTUC [12,13,14]. Recently, a relative new biomarker, the systemic immune-inflammation index (SII), based on neutrophile, lymphocyte, and platelet counts, has been developed and showed a high diagnostic value for the prognosis of urologic cancer [15]. The underlying mechanisms for the development of elevated SII in UTUC cancer patients and the subsequent increase in the poor survival of the disease remain unclear. However, these markers with respect to survival might be explained by the relationship among cancers to platelet, neutrophile, and lymphocyte levels. Platelets were reported to be a dangerous alliance of cancer cells and are a close engager in multiple processes of cancer metastasis [16]. Cancer cells can escape from immune cell cytotoxicity by platelet adhesion, which forms a protective cloak. Once tumor cells enter the blood, they immediately activate platelets to form a permissive microenvironment. Platelets save tumor cells from shear forces and NK cell assaults and secret chemokines to recruit myeloid cells. This causes the tumor cell platelet embolus to stop at the vascular wall. Then, platelet-derived growth factors confer a mesenchymal-like phenotype to tumor cells and open the capillary endothelium to expedite extravasation in distant organs. Eventually, platelet-secreted growth factors stimulate tumor cell proliferation to micro metastatic foci [17]. Neutrophils, the most abundant white blood cells in the circulation system, constitute a significant part of the tumor microenvironment. Tumor-associated neutrophils secrete cytokines and chemokines to exert an antitumor activity. Furthermore, neutrophiles stimulate immunosuppression, tumor growth, angiogenesis, and metastasis [18,19]. Therefore, the elevation of both platelets and neutrophiles reflect tumor aggressiveness. In contrast, lymphocytes are known to play an important role in cellular and humoral antitumor immune responses. Activated and proliferating lymphocytes play a role in cytotoxic cell death and inhibit tumor cell proliferation and migration by secreting cytokines such as interferon gamma (IFN-γ) and tumor necrosis factor alpha (TNF-α). Therefore, low lymphocyte count may reflect the impaired host immunosurveillance and represent an unfavorable prognostic factor for clinical outcomes in patients with tumors [20,21]. Considering these mechanisms, a higher SII representing higher platelet and neutrophile counts and lower lymphocytes may predict poor survival.

The prognostic significance of SII in UTUC or its relative utility when compared with other inflammatory indexes has not been fully investigated. Currently, only a few studies [6,22,23] have reported the predictive value of the SII as a predictor of UTUC treated by radical nephroureterectomy. In the present study, we retrospectively enrolled 376 patients and investigated the prognostic significance of SII in UTUC patients. Moreover, we compared the clinicopathologic effect on SII values with NLR and PLR.

## 2. Materials and Methods

### 2.1. Patient Selection

Patients who underwent either open or laparoscopic radical nephroureterectomy (RNU) with bladder cuff excision for non-metastatic UTUC between 2001 and 2013 at Kaohsiung Medical University Hospital were included. The present study was approved by the review board of our institution (KMUH-IRB-20120138, approved 12 July 2012). All patients signed the informed consent form. Furthermore, all patients routinely took blood examinations before surgical treatment. We retrospectively collected data regarding clinical parameters such as demographic characteristics, pathological features, oncologic follow-up, and the cause of death. We also excluded patients with immunological disorders, hematological disorders, bone marrow diseases, concurrent bladder tumor, neoadjuvant chemotherapy or radiotherapy and incomplete clinical information. Tumor stage was evaluated based on the 2002 American Joint Committee Cancer TNM system. The cases of all patients were reviewed by two pathologists and were re-classified as low- and high-grade based on the 2004 WHO grading system. The Chronic Kidney Disease Epidemiology Collaboration (CKD-EPI) creatinine-based formula [24] was used to evaluate patient renal function.

### 2.2. Treatment and Follow-Up

For low-risk tumors, the first cystoscopy was arranged at three months after operation. If negative, a subsequent cystoscopy was arranged nine months later. After that, annual cystoscopy was arranged for five years. Patients with high-risk tumor, cystoscopy and urinary cytology were suggested for testing every three months in the first two years and every six months in the next two years. After the fifth year, yearly follow-ups were arranged. For the first two years, computed tomography (CT) was arranged every six months. After that, yearly CT was suggested. European Association of Urology guidelines suggest that a single post-operative intravesical instillation of chemotherapy lowers the bladder cancer recurrence rate [2]. Despite level 1 evidence, only a small minority (42 patients, 11.2%) of patients received adjuvant single postoperative instillations of chemotherapy after RNU in our hospital.

### 2.3. Definitions of Inflammatory Indexes

The inflammatory indexes were calculated with preoperative inflammatory indicators with the following formulas: SII = platelet counts × neutrophile counts/lymphocyte counts; NLR = neutrophile counts/lymphocyte counts; and PLR = platelet counts/lymphocyte counts. The optimal cut-off value for metastasis-free survival (MFS), cancer-specific survival (CSS), and bladder recurrence-free survival (BRFS) were defined by creating a time-dependent receiver operating characteristic (ROC) curve with MFS, CSS, and BRFS, as the endpoint to yield the highest Youden index value (i.e., sensitivity + specificity −1), respectively. The Youden index can provide the optimal cut-off values from a continuous variable and offers the best for both sensitivity and specificity.

### 2.4. Statistical Analysis

The differences between categorical parameters were assessed using a χ^2^ or Fisher’s exact test. The Kaplan–Meier method was applied to estimate the effect of the three inflammatory indexes, SII, NLR, and PLR, as prognostic factors on metastasis-free survival (MFS), CSS, and bladder recurrence-free survival (BRFS). Survival rates were recorded from the day of RNU to metastatic progression, cancer-specific death, or the latest visit and bladder recurrence. Specifically, patients who died due to UTUC, including cancer progression and distant metastasis in the follow-up period, were defined to have had a cancer-specific death. Survival curves were compared using a log-rank test. Univariate and multivariate survival analyses were performed to identify risk factors for prognosis. Only significant prognostic factors in univariate analysis were included in the multivariate Cox proportional hazard model to identify independent predictors for MFS, CSS, and BRFS. The discrimination of the model was evaluated using the Harrell’s concordance index (C-index). The clinical benefit of SII was calculated using decision curve analysis (DCA). *p* < 0.05 was considered statistically significant. SPSS 20.0 (SPSS Inc., Chicago, IL, USA) and R version 4.1.1 (R Foundation for Statistical Computing, Vienna, Austria) were used for all statistical analyses.

## 3. Results

A total of 376 UTUC patients were included in the present study (Table 1). There were 146 (38.8%) male patients and 230 (61.2%) female patients. Female were dominant in our study cohort. The median age was 69.0 (IQR 15) years. The median follow-up time after surgery was 52.0 (IQR 16.7) months. There were 309 patients (82.2%) that underwent preoperative uretereorenoscopic (URS) exam and biopsy, 42 patients (11.2%) that underwent image-guided biopsy, and 25 patients (7.4%) with suspicious imaging features. Moreover, 101 patients (26.8%) had a history of bladder cancer, 146 patients (38.8%) had a pathologic pTa/pTis/pT1 stage, 68 (23.4%) had a pT2 stage, 114 (30.3%) a pT3 stage, and 28 (7.4%) had a pT4 stage. ROC analysis revealed that the optimal cut-off values of SII, NLR, and PLR for MFS and CSS were 460, 3.5, and 162 and 485, 2.9, and 150, respectively. The cut-off values of BRFS were 490, 3.5, and 160. Table 1, using the CSS cut-off values, shows that an elevated SII was associated with symptomatic hydronephrosis, bladder cancer history, advanced pathologic tumor stage, lymph node invasion, adjuvant chemotherapy and concomitant carcinoma in situ (CIS); high NLR was associated with older age, symptomatic hydronephrosis, hemodialysis status, anemia, multifocal tumor, advanced pathologic tumor stage, and adjuvant chemotherapy; and high PLR was associated with older age, anemia, advanced pathologic tumor stage, and adjuvant chemotherapy. These three inflammatory indexes were closely linked to each other. Patients who had advanced pathologic stage, lymph node invasion, and metastatic progression were candidates for adjuvant therapy. After obtaining the patients’ willingness and taking renal function into consideration, 71 cases received systemic chemotherapy and 30 cases received radiation therapy.

### Kaplan–Meier Analysis for MFS, CSS, and BRFS

Ninety-four patients (25.0%) from our study cohort experienced metastasis events (female, 58 patients; male, 36 patients). The 3- and 5-year MFS rates were 76.6% and 71.8%, respectively. Univariate analysis showed that bladder cancer history (*p* = 0.004), advanced CKD stage (*p* = 0.004), no hematuria (*p* < 0.001), symptomatic hydronephrosis (*p* < 0.001), ureteral tumors (*p* = 0.029), advanced pathological stage (*p* < 0.001), multifocal tumor (*p* = 0.013) lymph node invasion (*p* < 0.001), high tumor grade (*p* < 0.001), adjuvant chemotherapy (*p* < 0.001), elevated NLR (*p* < 0.001), elevated PLR (*p* < 0.001), and elevated SII (*p* < 0.001) were associated with worse MFS (Table 2). Symptomatic hydronephrosis (*p* = 0.002), ureteral tumors (*p* = 0.019), advanced pathological tumor stage (*p* < 0.001), high tumor grade (*p* = 0.039), adjuvant chemotherapy (*p* < 0.001), elevated NLR (*p* = 0.021), elevated PLR (*p* = 0.009), and elevated SII (*p* = 0.004) were independent risk factors for lower MFS in multivariate analysis. The Kaplan–Meier analysis indicated that patients exhibiting higher NLR, PLR, and SII showed significantly poor MFS rates (Figure 1a–c; log-rank, all *p* < 0.001). The C-index after inclusion of NLR, PLR and SII in MFS are 0.618, 0.655, and 0.670, respectively.

Sixty patients (15.9%) had cancer-specific mortality during follow-up (female, 38 patients; male, 22 patients). The 3- and 5-year CSS rates were 84.1% and 79.2%, respectively. Univariate analysis showed that obesity (*p* = 0.025), no hematuria (*p* < 0.001), symptomatic hydronephrosis (*p* < 0.001), advanced pathological stage (*p* < 0.001), lymph node invasion (*p* = 0.006), high tumor grade (*p* = 0.032), adjuvant chemotherapy (*p* < 0.001), elevated NLR (*p* < 0.001), elevated PLR (*p* = 0.009), and elevated SII (*p* < 0.001) were associated with worse CSS (Table 2). Obesity (*p* = 0.005), no hematuria (*p* = 0.034), symptomatic hydronephrosis (*p* < 0.001), advanced pathological tumor stage (*p* < 0.001), adjuvant chemotherapy (*p* < 0.001), elevated NLR (*p* = 0.009), elevated PLR (*p* = 0.046), and elevated SII (*p* = 0.014) were independent risk factors for lower CSS in multivariate analysis. Similarly, the Kaplan–Meier analysis revealed that patients exhibiting higher NLR, PLR, and SII showed significantly worse CSS rates (Figure 2a,b and Figure 3c). In CSS, the C-index of NLR, PLR and SII were 0.615, 0.676, and 0.677, respectively.

Additionally, 108 patients (28.7%) had bladder recurrence during follow-up (female, 52 patients; male, 56 patients). The 3- and 5-year BRFS rates were 78.6% and 63.5%, respectively. Univariate analysis showed that male gender (*p* = 0.001), bladder cancer history (*p* = 0.001), pT4 pathological stage (*p* = 0.011), high tumor grade (*p* = 0.020), adjuvant chemotherapy (*p* < 0.001), and elevated SII (*p* = 0.026) were associated with worse BRFS (Table 2). Male gender (*p* = 0.001), bladder cancer history (*p* <0.001), pT4 pathological stage (*p* = 0.032), high tumor grade (*p* = 0.010), adjuvant chemotherapy (*p* < 0.001) and elevated SII (*p* = 0.016) were independent risk factors for lower BRFS in multivariate analysis. Adjuvant single postoperative instillations of chemotherapy after RNU can decrease the bladder recurrence rate in univariate analysis but fail to achieve statistic difference in multivariate analysis. Further, the Kaplan–Meier analysis indicated that patients with higher SII showed significantly worse BRFS rates (Figure 3c). Higher NLR and PLR were not associated with BRFS (Figure 3a,b). A further analysis was performed to determine the recurrence tumor’s pathological stage when diagnosed (Appendix A). Among patients with bladder recurrence after nephroureterectomy, 70 patients (64.8%) had elevated SII values. We also noted that elevated of SII was associated with a higher rate of preoperative hydronephrosis (*p* = 0.018), bladder cancer history *(p* = 0.044), and high-grade recurrence tumor (*p* = 0.026). Among 41 patients who received adjuvant single postoperative instillations of chemotherapy, only seven patients (16.7%) had bladder recurrence. The C-index after considering the NLR, PLR, and SII are 0.640, 0.688, and 0.710, respectively. On DCA, SII did not improve the net benefit (Appendix A).

## 4. Discussion

To our knowledge, this is the first study that investigated the prognostic significance of SII on BRFS in UTUC patients. Moreover, we compared the significance of preoperative NLR and PLR with SII in UTUC patients. We demonstrated that elevated SII was not only associated with worse oncologic outcomes, but also predicted bladder recurrence. Our study has potentially crucial clinical implications. By performing the simple routine preoperative blood examinations, we can identify patients who have a higher risk for poor survival rates. SII was a more superior prognostic predictor of bladder recurrence than NLR and PLR. In the current study cohort, elevated SII was noted in 56% of patients, and one-third of these patients suffered from a bladder recurrence. A careful postoperative follow-up should be performed in these patients.

A recent meta-analysis [15] analyzed the prognostic value of SII in urologic cancer. It included 14 published papers with 3074 patients and showed that UC patients with high SII had a worse prognosis for overall survival. Furthermore, it concluded that SII could be a simple and cost-effective prognostic indicator. In UTUC, three previous studies have investigated the prognostic significance of SII. Jan et al. [22] combined SII and monocyte-to-lymphocyte ratio (MLR) to predict disease progression and survival. They concluded that using both high SII and high MLR were significantly more likely to predict non-organ-confined disease and poor survival outcomes than single indicator use. Zheng et al. [23] used the combination of simultaneous SII and prognostic nutritional index (PNI) and showed a powerful independent risk factor for overall survival, CSS, and recurrence-free survival. The combination had the largest area under the curve compared to that for SII or PNI alone and other clinical factors, indicating its advantage in predicting survival. A large multi-institutional cohort from the international UTUC collaboration study that included 2492 cases with RNU showed that altered SII was significantly associated with higher pathologic stages and worse survival outcomes [6]. Additionally, the decision curve analysis was used to evaluate the clinical benefit of SII. However, the preoperative SII appears to have relatively limited incremental additive values for clinical decision-making. Unlike the previous two studies [22,23] that combined other parameters, our results were based on the SII alone and show value in providing insight into preoperative prognostic prediction (Appendix A). Moreover, we demonstrated that the SII not only reflects survival prediction but also pathologic surgical outcomes which is more relevant to clinical practice [6]. Among patients with bladder recurrence after RNU, SII elevation was associated with high grade recurrence tumors. Therefore, a stricter and more frequent follow-up should be advised.

An increasing number of studies have proved that blood-based inflammatory markers are emerging predictors for prognosis of various tumors [10,11,12,13,14,15,16]. The current immune checkpoint blockade therapy has shown promising anti-tumor outcomes in many malignancies, including UTUC [25]. There have been studies reporting SII significance in the survival of patients undergoing immunotherapy [26]. Aside from the baseline biomarkers, we believe that the alteration of biomarkers during immunotherapy treatment would provide greater information and indicate an immediate effect from the immunotherapy treatment itself [27]. Shang et al. [27] reported that the changes of NLR as prognostic biomarkers for patients with pancreatic cancer treated with immunotherapy in multivariate analysis. Lalani et al. [28] showed that relative over 25% NLR change from baseline to six weeks after immunotherapy was associated with reduced objective response rate and an independent prognostic factor for progression free survival (*p* < 0.001) and overall survival (*p* = 0.004), whereas a 25% decrease in NLR was associated with better outcomes in metastatic renal cell carcinoma. The dynamic NLR changes may imply the effects of immunotherapy on the immune system, regardless of baseline values. Because of the inexpensive and readily available feature of NLR, it is quite convenient to calculate and monitor biomarker changes during clinical courses.

EAU guidelines suggest that a single post-operative intravesical instillation of chemotherapy 2–10 d after surgery reduces the risk of bladder tumor recurrence within the initial years [2]. We had also assessed the efficacy of prophylactic intravesical chemotherapy for primary upper urinary tract urothelial cancer after surgery and showed that the instillation of epirubicin or mitomycin C appears to be well tolerated and effective for preventing bladder recurrence and prolonging time to first bladder recurrence [29]. We fully agreed on the issue of the potential benefits of single post-operative intravesical therapy. Meanwhile, there is a lack of consensus regarding the prognostic significance of different approaches to the bladder cuff at the time of surgery. We also compared the oncologic outcomes following nephroureterectomy using three different methods: intravesical incision, extravesical incision, and transurethral incision (TUI), of managing the bladder cuff [30]. Our results showed no differences in bladder recurrence and cancer-specific survival among the three groups, but the TUI method seems to have a lower, although not significantly different, recurrence rate (intravesical, extravesical, and TUI techniques, bladder recurrence developed in, respectively, 23.5%, 24.0%, and 17.6% cases (*p* = 0.485); local retroperitoneal recurrence in 7.4%, 7.8%, and 5.5% (*p* = 0.798); contralateral recurrence in 4.9%, 3.9%, and 2.2% (*p* = 0.632); and distant metastasis in 7.4%, 10.4%, and 5.5% (*p* = 0.564). Therefore, more doctors preferred the TUI method in our hospital. Because the TUI method does not suture the bladder cuff opening, we found that the post-operative extravasation rate is much higher than those with bladder suturing. As a result, immediate intravesical instillation has not become a common practice in recent years in our hospital.

Clearly, the female dominant distributions of UTUC in Taiwan differ from those in other regions in the world [4,5,22,29,30]. Chen et al. [31] attributes the progressive increase in the high incidence of UTUC, especially among women, in part to the systematic replacement of traditionally used Chinese herbs with aristolochic acid based on aristolactam-DNA adducts and TP53 mutations, which are identical to those observed in UTUC associated with Balkan endemic nephropathy. One of the culturally based reasons is that women consume a special tonic supplement involving herbal medicines daily for at least one month after each pregnancy [32]. The exposure to aristolochic acid contributes significantly to the high incidence of UTUC in women in Taiwan.

This study had several limitations. First, our results were based on a single-center experience. Second, the fact that different surgeons may have their own preferences regarding surgical approach could represent a selection bias. Third, we could not all possible confounding maneuver which may influence the SII level. Fourth, Biomarkers were assessed only preoperatively at a single time point; their variability over time and their response to the treatment have not been assessed. Fifth, due to the very low proportion of patients that received intravesical chemotherapy after nephroureterectomy, the results presented cannot be generalized to current practice. Future larger studies are warranted to confirm our findings.

## 5. Conclusions

We found that high preoperative NLR, PLR, and SII were independent predictive factors for both MFS and CSS in UTUC patients. Among the factors, only elevated SII can predict bladder recurrence. Therefore, close bladder monitoring during follow-up is recommended.

## Figures and Tables

**Figure 1 jcm-10-05273-f001:**
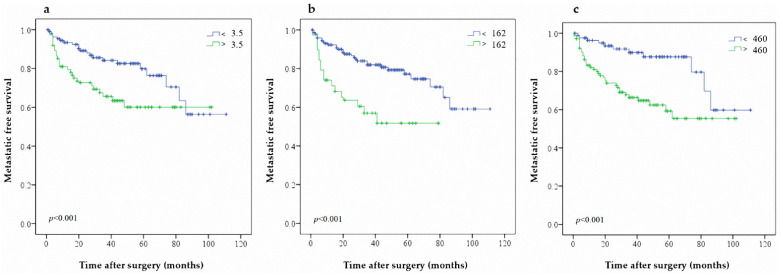
Predictive value of (**a**) NLR (**b**) PLR and (**c**) SII on MFS in patients with UTUC after RNU.

**Figure 2 jcm-10-05273-f002:**
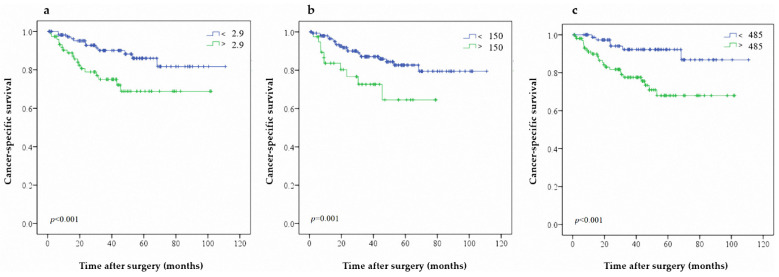
Predictive value of (**a**) NLR (**b**) PLR and (**c**) SII on CSS in patients with UTUC after RNU.

**Figure 3 jcm-10-05273-f003:**
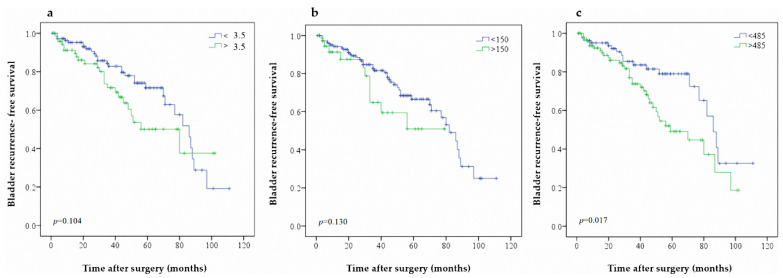
Predictive value of (**a**) NLR (**b**) PLR and (**c**) SII on BRFS in patients with UTUC after RNU.

**Table 1 jcm-10-05273-t001:** Associations of NLR, PLR, SII and clinicopathologic characteristics in 376 patients treated with radical nephroureterectomy for upper tract urothelial carcinoma.

Parameters	All		NLR			PLR			SII	
		>2.9	≤2.9	*p*-Value	>150	≤150	*p*-Value	>485	≤485	*p*-Value
	*n* (%)	*n* (%)	*n* (%)		*n* (%)	*n* (%)		*n* (%)	*n* (%)	
Age				**0.006**			**<0.001**			0.972
≤70	208 (55.3)	70 (46.7)	138 (61.1)		28 (35.9)	180 (60.4)		116 (55.2)	92 (55.4)	
>70	168 (44.7)	80 (53.1)	88 (38.9)		50 (64.2)	118 (39.6)		94 (44.8)	74 (44.6)	
Gender				0.214			0.263			0.601
Male	146 (38.8)	64 (42.7)	82 (36.3)		26 (33.3)	120 (40.3)		84 (40.0)	62 (37.3)	
Female	230 (61.2)	86 (57.3)	144 (63.7)		52 (66.7)	178 (59.7)		126 (60.0)	104 (62.7)	
Hematuria				0.296			0.196			0.681
No	83 (22.1)	29 (19.3)	54 (23.9)		13 (16.7)	70 (23.5)		48 (22.9)	35 (21.1)	
Yes	293 (77.9)	121 (80.7)	172 (76.1)		65 (83.3)	228 (76.5)		162 (77.1)	131 (78.9)	
Symptomatic hydronephrosis				**0.014**			0.572			**0.021**
No	330 (87.8)	124 (82.7)	206 (91.2)		67 (85.9)	263 (88.3)		177 (84.3)	153 (92.2)	
Yes	46 (12.2)	26 (17.3)	20 (8.8)		11 (14.1)	35 (11.7)		33 (15.7)	13 (7.8)	
Bladder cancer history				0.263			0.989			**0.025**
No	275 (73.1)	105 (70.0)	170 (75.2)		57 (73.1)	218 (73.2)		144 (68.6)	131 (78.9)	
Yes	101 (26.9)	45 (30.0)	56 (24.8)		21 (26.9)	80 (26.8)		66 (31.4)	35 (21.1)	
BMI				0.056			0.096			0.462
>27	62 (16.5)	18 (12.0)	44 (19.5)		8 (10.3)	54 (18.1)		32 (15.2)	30 (18.1)	
≤27	314 (83.5)	132 (88.0)	182 (80.5)		70 (89.7)	244 (81.9)		178 (84.8)	136 (81.9)	
ESRD				**0.044**			0.479			0.151
No	300 (79.8)	112 (74.7)	188 (83.2)		60 (76.9)	240 (80.5)		162 (77.1)	138 (83.1)	
Yes	76 (20.2)	38 (25.3)	38 (16.8)		18 (23.1)	58 (19.5)		48 (22.9)	28 (16.9)	
Anemia				**0.016**			**<0.001**			0.127
No	138 (36.7)	44 (29.3)	94 (41.6)		14 (17.9)	124 (41.6)		70 (33.3)	68 (41.0)	
Yes	238 (63.3)	106 (70.7)	132 (58.4)		64 (82.1)	174 (58.4)		140 (66.7)	98 (59.0)	
CKD stage				0.053			0.223			0.699
Stage 1	36 (9.6)	6 (4.0)	30 (13.3)		4 (5.1)	32 (10.7)		18 (8.6)	18 (10.8)	
Stage 2	78 (20.7)	32 (21.3)	46 (20.4)		18 (23.1)	60 (20.1)		44 (21.0)	34 (20.5)	
Stage 3	154 (41.0)	64 (42.7)	90 (39.8)		28 (35.9)	126 (42.3)		84 (40.0)	70 (42.2)	
Stage 4	24 (6.4)	10 (6.7)	14 (6.2)		8 (10.3)	16 (5.4)		12 (5.7)	12 (7.2)	
Stage 5	84 (22.3)	38 (25.3)	46 (20.4)		20 (25.6)	64 (21.5)		52 (24.8)	32 (19.3)	
Advanced CKD stage (Stage 4, 5)	108 (28.7)	48 (32.0)	60 (26.5)	0.253	28 (35.9)	80 (26.8)	0.116	64 (30.5)	44 (26.5)	0.398
Type of operation				0.304			0.712			0.093
Open	244 (64.9)	48 (32.0)	84 (37.2)		52 (66.7)	192 (66.4)		144 (68.6)	100 (60.2)	
Laparoscopic	132 (35.1)	102 (68.0)	142 (62.8)		26 (33.3)	106 (35.6)		66 (31.4)	66 (39.8)	
Tumor location				0.910			0.131			0.445
Pyelocaliceal	170 (45.2)	68 (45.3)	102 (45.1)		42 (53.8)	128 (43.0)		98 (46.7)	72 (43.4)	
Ureteral	132 (35.1)	54 (36.0)	78 (34.5)		26 (33.3)	106 (35.6)		68 (32.4)	64 (38.6)	
Both	74 (19.7)	28 (18.7)	46 (20.4)		10 (12.8)	64 (21.5)		44 (21.0)	30 (18.1)	
Multifocality				**0.021**			0.568			0.099
Single	298 (79.3)	110 (73.3)	188 (83.2)		60 (76.9)	238 (79.9)		160 (76.2)	138 (83.1)	
Multiple	78 (20.7)	40 (26.7)	38 (16.8)		18 (23.1)	60 (20.1)		50 (23.8)	28 (16.9)	
Pathologic T stage				**0.001**			**0.018**			**0.017**
pTa/pTis/pT1	146 (38.8)	46 (30.7)	100 (44.2)		26 (33.3)	120 (40.3)		70 (33.3)	76 (45.8)	
pT2	68 (23.4)	38 (25.3)	50 (22.1)		20 (25.6)	68 (22.8)		52 (24.8)	36 (21.7)	
pT3	114 (30.3)	46 (30.7)	68 (30.1)		20 (25.6)	94 (31.5)		66 (31.4)	48 (28.9)	
pT4	28 (7.4)	20 (13.3)	8 (3.5)		12 (15.4)	16 (5.4)		22 (10.5)	6 (3.6)	
Advanced pT stage (pT3, pT4)				**0.042**			0.505			0.063
No	234 (62.2)	84 (56.0)	150 (66.4)		46 (59.0)	188 (63.1)		122 (58.1)	112 (67.5)	
Yes	142 (37.8)	66 (44.0)	76 (33.6)		32 (41.0)	110 (36.9)		88 (41.9)	54 (32.5)	
Pathologic N stage				0.659			0.479			**0.003**
pN0	68 (18.1)	23 (15.3)	36 (15.9)		22 (22.4)	42 (14.1)		31 (14.8)	35 (21.1)	
pNx	232 (61.7)	95 (63.3)	146 (64.6)		58 (59.2)	198 (66.4)		125 (59.5)	109 (65.7)	
pN+	76 (20.2)	32 (21.3)	44 (19.5)		18 (23.1)	58 (19.5)		54 (25.7)	22 (13.3)	
Grade				0.257			0.941			0.527
Low	76 (20.2)	26 (17.3)	50 (22.1)		16 (20.5)	60 (20.1)		40 (19.0)	36 (21.7)	
High	300 (79.8)	124 (82.7)	176 (77.9)		62 (79.5)	238 (79.9)		170 (81.0)	130 (78.3)	
Concomitant CIS				0.323			0.235			**0.028**
No	331 (88.0)	129 (86.0)	202 (89.4)		62 (79.5)	169 (85.4)		178 (84.8)	153 (92.2)	
Yes	45 (12.0)	21 (14.0)	24 (10.6)		16 (20.5)	29 (14.6)		32 (15.2)	13 (7.8)	
Adjuvant chemotherapy				**0.015**			**0.009**			**<0.001**
No	308 (81.9)	114 (76.0)	194 (85.8)		56 (71.8)	252 (84.6)		158 (75.2)	150 (90.4)	
Yes	68 (18.1)	36 (24.0)	32 (14.2)		22 (28.2)	46 (15.4)		52 (24.8)	16 (9.6)	
NLR							**<0.001**			**<0.001**
>2.9	-	-	-		76 (97.4)	74 (24.8)		140 (66.7)	10 (6.0)	
≤2.9	-	-	-		2 (2.6)	224 (75.2)		70 (33.3)	156 (94.0)	
PLR				**<0.001**						**<0.001**
>150	78 (20.7)	76 (50.7)	2 (0.9)		-	-		78 (37.1)	0 (0.0)	
≤150	298 (79.3)	74 (49.3)	224 (99.1)		-	-		132 (62.9)	166 (100.0)	
SII				**<0.001**			**<0.001**			
>485	210 (55.9)	140 (93.3)	70 (31.0)		78 (100.0)	132 (44.3)		-	-	
≤485	166 (44.1)	10 (6.7)	156 (69.0)		0 (0.0)	166 (55.7)		-	-	

CKD = chronic kidney disease; ECOG = Eastern Cooperative Oncology Group; ESRD = end stage renal disease; NLR = neutrophile-lymphocyte ratio; PLR = Platelet-lymphocyte ratio; SII = systemic immune-inflammation index. Statistically significant values are marked with Bold.

**Table 2 jcm-10-05273-t002:** Univariate and multivariate analyses predicting CSS, MFS and BRFS in patients (*n* = 376) with UTUC after RNU.

Parameters	MFS				CSS				BRFS			
	Univariate Analysis	*p*	Multivariate Analysis	*p*	Univariate Analysis	*p*	Multivariate Analysis	*p*	Univariate Analysis	*p*	Multivariate Analysis	*p*
	HR (95%CI)		HR (95%CI)		HR (95%CI)		HR (95%CI)		HR (95%CI)		HR (95%CI)	
Age (Years)												
Over 70 years	1.1 (0.7–1.8)	0.632			1.3 (0.7–2.2)	0.366			1.2 (0.8–1.9)	0.391		
Gender												
Male vs. Female	1.0 (0.9–1.2)	0.509			1.2 (0.9–1.6)	0.329			2.2 (1.4–3.4)	**0.001**	2.4 (1.5–4.0)	**0.001**
Bladder cancer history												
Yes vs. No	1.3 (1.1–1.5)	**0.004**	1.2 (0.9–1.4)	0.586	1.2 (0.9–1.5)	0.126			1.8 (1.5–2.1)	**<0.001**	1.6 (1.4–1.9)	**<0.001**
BMI (Kg/m^2^)												
Over 27 Kg/m^2^	1.7 (0.9–3.0)	0.078			2.8 (1.0–7.3)	**0.025**	6.1 (1.7–21.1)	**0.005**	1.1 (0.7–1.8)	0.579		
Advanced CKD (Stage 4, 5)												
Yes vs. No	2.0 (1.2–3.2)	**0.004**	1.2 (0.5–2.7)	0.686	1.6 (0.9–2.9)	0.103			1.2 (0.7–2.0)	0.453		
Hematuria												
Yes vs. No	0.7 (0.6–0.9)	**<0.001**	0.9 (0.7–1.1)	0.162	0.6 (0.5–0.8)	**<0.001**	0.8 (0.6–0.9)	**0.034**	0.9 (0.8–1.1)	0.385		
Symptomatic hydronephrosis												
Yes vs. No	1.5 (1.2–1.8)	**<0.001**	1.4 (1.1–1.7)	**0.002**	1.9 (1.5–2.4)	**<0.001**	1.6 (1.2–2.1)	**<0.001**	0.7 (0.7–1.1)	0.051		
Type of operation												
Laparoscopic vs. open	0.9 (0.6–1.5)	0.803			0.6 (0.3–1.2)	0.135			0.7 (0.4–1.1)	0.158		
Tumor location												
Ureteral vs. Pyelocaliceal	2.5 (1.1–5.7)	**0.029**	4.2 (1.3–13.7)	**0.019**	1.2 (0.7–2.3)	0.058			1.0 (0.6–1.7)	0.967		
Both vs. Ureteral	1.5 (0.5–4.4)	0.318	1.2 (0.6–2.6)	0.211	1.2 (0.6–2.5)	0.683			1.4 (0.8–2.7)	0.238		
Both vs. Pyelocaliceal	1.6 (0.6–4.2)	0.430	2.8 (0.8–9.0)	0.095	1.4 (0.7–2.9)	0.342			1.5 (0.8–2.6)	0.204		
Multifocality												
Multiple vs. Single	2.0 (1.2–3.4)	**0.013**	1.6 (0.7–3.9)	0.285	1.5 (0.8–2.8)	0.217			1.1 (0.7–1.9)	0.654		
Pathologic T stage												
pT2 vs. pTa/pTis/pT1	3.0 (1.3–7.0)	**0.008**	1.2 (0.2–6.8)	0.185	1.7 (0.6–4.8)	**0.289**	1.4 (0.4–4.1)	0.593	1.0 (0.6–1.7)	0.924	1.2 (0.6–2.2)	0.683
pT3 vs. pTa/pTis/pT1	12.2 (5.8–25.7)	**<0.001**	1.8 (0.9–3.5)	0.177	7.3 (3.2–16.6)	**<0.001**	2.2 (1.4–3.5)	**0.001**	1.1 (0.6–1.8)	0.803	1.4 (0.9–2.1)	0.133
pT4 vs. pTa/pTis/pT1	13.6 (5.1–36.3)	**<0.001**	2.6 (1.5–4.7)	**<0.001**	9.6 (3.3–27.4)	**<0.001**	1.6 (1.1–2.4)	**0.032**	4.2 (1.1–16.4)	**0.011**	1.8 (1.1–3.2)	**0.032**
Pathologic N stage												
pN+ vs. pN0/pNx	3.8 (2.2–6.4)	**<0.001**	1.2 (0.5–2.9)	0.640	2.3 (1.3–4.3)	**0.006**	1.3 (0.6–3.0)	0.467	1.1 (0.7–1.7)	0.604		
Grade												
High vs. Low	5.7 (2.2–15.0)	**<0.001**	4.4 (1.1–17.9)	**0.039**	2.3 (1.3–4.2)	**0.032**	1.1 (0.4–3.6)	0.830	1.9 (1.1–3.1)	**0.020**	2.5 (1.4–4.4)	**0.010**
Concomitant CIS												
Yes vs. No	1.4 (1.2–1.6)	**0.002**	1.3 (1.3–2.2)	**0.018**	1.5 (1.2–3.6)	**<0.001**	1.4 (1.2–2.4)	**0.038**	1.1 (0.8–1.3)	0.791		
Adjuvant chemotherapy												
Yes vs. No	43.8 (20.6–93.3)	**<0.001**	40.7 (16.3–101.7)	**<0.001**	8.9 (4.8–16.4)	**<0.001**	6.6 (3.2–13.7)	**<0.001**	4.3 (2.5–7.4)	**<0.001**	4.8 (2.7–8.5)	**<0.001**
Post-operative installation chemotherapy												
Yes vs. No	0.9 (0.8–1.2)	0.785			1.0 (0.6–1.3)	0.788			1.1 (1.0–1.2)	**0.028**	1.0 (0.9–1.1)	0.678
NLR	(>3.5 vs. ≤3.5)				(>2.9 vs. ≤2.9)				(>3.5 vs. ≤3.5)			
	2.3 (1.4–3.7)	**<0.001**	2.2 (1.1–4.3)	**0.021**	2.7 (1.5–4.7)	**<0.001**	2.3 (1.2–4.3)	**0.009**	1.3 (0.8–2.0)	0.253		
PLR	(>162 vs. ≤162)				(>150 vs. ≤150)				(>160 vs. ≤160)			
	3.1 (1.8–5.2)	**<0.001**	3.6 (1.4–9.4)	**0.009**	2.2 (1.2–4.1)	**0.009**	2.0 (1.1–4.0)	**0.046**	1.0 (0.7–1.5)	0.910		
SII	(>460 vs. ≤460)				(>485 vs. ≤485)				(>490 vs. ≤490)			
	3.4 (2.0–5.8)	**<0.001**	2.9 (1.4–6.1)	**0.004**	3.8 (2.0–7.4)	**<0.001**	3.3 (1.3–8.5)	**0.014**	1.7 (1.1–2.7)	**0.026**	1.7 (1.1–2.8)	**0.016**

CKD = chronic kidney disease; ECOG = Eastern Cooperative Oncology Group; ESRD = end stage renal disease; NLR = neutrophile-lymphocyte ratio; PLR = Platelet-lymphocyte ratio; SII = systemic immune-inflammation index. Statistically significant values are marked with Bold.

## Data Availability

The data presented in this study are available on request from the corresponding author.

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
