# Peer review of "The Predictive Value of Systemic Immune-Inflammation Index on Bladder Recurrence on Upper Tract Urothelial Carcinoma Outcomes after Radical Nephroureterectomy"

_jcm, 2021, doi:10.3390/jcm10225273_

Round 1

Reviewer 1 Report

The authors conducted a retrospective study aimed to assess the prognostic significance of pre-treatment lymphocyte-related systemic inflammatory biomarkers in upper tract urothelial carcinoma (UTUC) patients. The authors reported that the NLR, PLR, and SII were independent predictive factors for both MFS and CSS in UTUC patients; while only elevated SII can predict bladder recurrence.

The outcomes of this study may pose an interest and became the basis for further research.

The purpose, methods, results, and data interpretation of the study are mainly informative and satisfactory. However, the authors should address some questions and problems that appeared during the manuscript review.

  • Why did not the authors analyze the combination of reported biomarkers? Its combination might improve its net benefit as single biomarkers.

Results

  • Please report the distribution of concomitant CIS between the groups.
  • Consider reporting all continuous variables as median (IQR).
  • Consider reducing the number of co-variants in multivariable analysis. For 94 patients who experienced metastases, 15 co-variants can lead to underpowered results. For 60 patients who died from UTUC, 10 co-variants can lead to underpowered results.
  • The discriminative ability of the models before and after the inclusion of NLR, PLR, and SII should be tested using C-index.
  • It is not clear if the addition of NLR, PLR, or SII improved the net benefit compared to the standard model based on clinicopathological features. Consider performing a decision curve analysis to evaluate the clinical net benefit using models with and without systemic inflammatory biomarkers for predicting survival probabilities (MFS, CSS, BRFS).

Discussion

  • Were patients with immunological or hematological disorders excluded from the present study?
  • Biomarkers assessed only preoperatively at a single time point, while their variability over time and their response to the treatment have not been assessed. Consider reporting this fact among the Limitations of your study.
  • It is also promising to study these biomarkers reflecting inflammation in the context of immunotherapy. Please comment on this aspect in the Discussion section.
  • The third paragraph of the Discussion section seems to be out of scope for the present study. Please, instead of discussing different RNU approaches, consider discussing the clinical applicability of your results.

Reviewer 2 Report

The authors present a retrospective single-center evaluation of the prognostic value of inflammatory markers prior to radical nephro-ureterectomy (RNU) in non-metastatic upper tract urothelial carcinomas.

Although all 3 indexes correlated with both MFS and CSS, only SII predicted bladder recurrence. This is strongly stressed by the authors both in the text and especially in the title.

Nevertheless, this no corollary like a new proposed scheme for surveillance. There should be further analysis to determine when do bladder recurrences occur and its pathological stage when diagnosed. For instance, if most bladder recurrences were non-invasive low-grade urothelial carcinomas that only required TUR, then there is no clinical significance to this result. On the contrary, if most recurrences or high-grade or invasive, then a more stric and frequent follow-up should be advised.

Another important issue is the the very low proportion of patients that received intravesical chemotherapy after RNU. Had almost all of them received it, and the bladder recurrence rate would have been significantly lower (but perhaps of higher severity). Therefore, the results presented cannot be generalized to current practice. This must be stressed in the discussion section as a serious limitation of the study.

Another question is related to which patients were selected to receive adjuvant chemotherapy (all patients with advanced pathological stage and normal renal function or was it left at the assistant urologist's discretion?). Please present data about this in the Results section.

Reviewer 3 Report

Ratios are nowaday frequently reintroduced as prognostic, easy accessibly and cost effective, tools. SII is a derivate potentially reflecting at time 0 patient immune and inflammatory background. The results you described are intriguing as the latter could help the clinical practice predicting high risk of bladder recurrence in this challenging clinical scenario. However other ratios are well described and investigated in urological cancer and less studied in UTUC.

However some data are missing:

  • What kind of endourological or percoutaneous management patients underwent before nephroureterectomy.
  • No patient underwent a conservative management attempt? or topic therapy protocol?
  • bladder status before surgery (all patient had a negative cistoscopy before surgery?).  Only 101 patietns had NMIBC history (pT stage?, previous intravescical therapy?). You excluded only patients with MIBC ?
  • SII was higher in patients bladder cancer history, this could be influenced by treatments and controls workout with potentially confounding proinflammative manouvre(repeated cistoscopy, BCG, catheters etc)
  • Hydronephrosis/haematuria status at time of diagnosis and at time of surgery /Stenting status.

  • You didn't find any difference in oncological outcomes and bladder recurrence in population undergoing open vs miniinvasive nephroureterectomy; do you think this could be related to your expertize in managing bladder cuff? Surgical approach could represent a selection bias.
  • Did you find a protective role of single post-operative chemoterapy instillation  on bladder recurrence in this sub-population?
  • Can you explain the Female/Male ratio for this malignancy? Taiwanese incidence among women seems to be the highest; any ambiental or genetic factor? (arsenic exposure, analgesic, chinese herb use etc)

Round 2

Reviewer 1 Report

Thank the authors for their response to my comments. However, there are still some issues that might be improved. 

  1. Results on C-index and decision curve analyses from file with answers to the reviewer should be implemented in the main text. Consider showing figures with DCA. 

Reviewer 2 Report

The authors have adequately addressed the issues raised and significantly improved the manuscript.

Nevertheless, in line 319, I would remove the expression "much lower recurrence rate" as the difference is not statistically significant. I would prefer "lower, although not significantly different, recurrence rate".
